# A Comparative Endocrine Trans-Differentiation Approach to Pancreatic Ductal Adenocarcinoma Cells with Different EMT Phenotypes Identifies Quasi-Mesenchymal Tumor Cells as Those with Highest Plasticity

**DOI:** 10.3390/cancers13184663

**Published:** 2021-09-17

**Authors:** Paula M. Schmidtlein, Clara Volz, Rüdiger Braun, Isabel Thürling, Olha Lapshyna, Ulrich F. Wellner, Björn Konukiewitz, Hendrik Lehnert, Jens-Uwe Marquardt, Hendrik Ungefroren

**Affiliations:** 1First Department of Medicine, Campus Lübeck, University Hospital Schleswig-Holstein, D-23538 Lübeck, Germany; paula.schmidtlein@student.uni-luebeck.de (P.M.S.); clara.volz@student.uni-luebeck.de (C.V.); isabel.thuerling@student.uni-luebeck.de (I.T.); Jens.Marquardt@uksh.de (J.-U.M.); 2Clinic for Surgery, Campus Lübeck, University Hospital Schleswig-Holstein, D-23538 Lübeck, Germany; ruediger.braun@uksh.de (R.B.); olha.lapshyna@uksh.de (O.L.); ulrich.wellner@uksh.de (U.F.W.); 3Institute of Pathology, Campus Kiel, University Hospital Schleswig-Holstein, D-24105 Kiel, Germany; Bjoern.Konukiewitz@uksh.de; 4University of Salzburg, A-5020 Salzburg, Austria; hendrik.lehnert@sbg.ac.at

**Keywords:** pancreatic ductal adenocarcinoma, plasticity, quasi-mesenchymal, epithelial, trans-differentiation, endocrine, pancreatic β cell, insulin

## Abstract

**Simple Summary:**

Pancreatic ductal adenocarcinoma (PDAC) is one of the deadliest cancer types with the quasi-mesenchymal (QM) subtype of PDAC having the worst prognosis. De-differentiation of the ductal tumor cells to a mesenchymal phenotype occurs as a result of epithelial–mesenchymal transition (EMT), a process associated with the acquisition of stem cell traits. While QM tumor cells are highly metastatic and drug-resistant, their increased plasticity opens a window of opportunity for trans-differentiation into non-malignant pancreatic cells. In this study we compared established PDAC-derived cell lines of either epithelial (E) or QM phenotype for their potential to be differentiated to pancreatic endocrine cells. We found that QM cells responded more strongly than E cells with transcriptional activation of a pancreatic progenitor or pancreatic β cell-specific program. Our results bear strong implications for a novel type of targeted therapy, namely EMT-based trans-differentiation of highly metastatic PDAC cells in vivo to non-malignant endocrine cells.

**Abstract:**

Pancreatic ductal adenocarcinoma (PDAC) is one of the most aggressive and therapy-resistant cancer types which is largely due to tumor heterogeneity, cancer cell de-differentiation, and early metastatic spread. The major molecular subtypes of PDAC are designated classical/epithelial (E) and quasi-mesenchymal (QM) subtypes, with the latter having the worst prognosis. Epithelial–mesenchymal transition (EMT) and the reverse process, mesenchymal-epithelial transition (MET), are involved in regulating invasion/metastasis and stem cell generation in cancer cells but also early pancreatic endocrine differentiation or de-differentiation of adult pancreatic islet cells in vitro, suggesting that pancreatic ductal exocrine and endocrine cells share common EMT programs. Using a panel of PDAC-derived cell lines classified by epithelial/mesenchymal expression as either E or QM, we compared their trans-differentiation (TD) potential to endocrine progenitor or β cell-like cells since studies with human pancreatic cancer cells for possible future TD therapy in PDAC patients are not available so far. We observed that QM cell lines responded strongly to TD culture using as inducers 5′-aza-2′-deoxycytidine or growth factors/cytokines, while their E counterparts were refractory or showed only a weak response. Moreover, the gain of plasticity was associated with a decrease in proliferative and migratory activities and was directly related to epigenetic changes acquired during selection of a metastatic phenotype as revealed by TD experiments using the paired isogenic COLO 357-L3.6pl model. Our data indicate that a QM phenotype in PDAC coincides with increased plasticity and heightened trans-differentiation potential to activate a pancreatic β cell-specific transcriptional program. We strongly assume that this specific biological feature has potential to be exploited clinically in TD-based therapy to convert metastatic PDAC cells into less malignant or even benign cells.

## 1. Introduction

Pancreatic ductal adenocarcinoma (PDAC) is a highly aggressive disease with an extremely poor prognosis (5-year survival ~7.7%) [1]. Its incidence is rising steeply, and it is expected to become the second leading cause of cancer-related death in 2030. At the time of diagnosis, the vast majority of patients present with locally advanced or metastatic disease. Surgery is currently the only curative option, however, only ~20% of patients are amenable to this treatment option and even those that have undergone successful R_0_ resection develop liver metastases shortly after surgery [2]. A series of genomic and transcriptomic studies have demonstrated that human PDAC is not a homogeneous disease but is composed of subtypes with different functional behaviors in preclinical models and differences in clinical studies with respect to survival and response to drug treatment [3,4]. The major molecular subtypes consistently found in PDAC are designated classical/epithelial (E) and quasi-mesenchymal (QM)/squamous/basal-like. However, single-cell transcriptomic analysis has revealed an unanticipated high heterogeneity of pancreatic cancers and demonstrated that QM cells are even present in E tumors [5]. These cells have been associated with systemic metastasis, escape from standard therapy and tumor recurrence [4,6].

Molecular subtype specification is thought to depend on high phenotypic variability in tumor cells. A driving force of this phenomenon is epithelial–mesenchymal transition (EMT) [7], a developmental program characterized by loss of epithelial characteristics, i.e., downregulation of E-cadherin (ECAD) expression and upregulation of mesenchymal markers like vimentin (VIM) in a Snail or Slug-dependent manner [8]. The group of EMT transcription factors also includes Zeb1 and Twist, which are particularly important in PDAC [9]. EMT reprogramming in tumors is closely associated with the generation of cancer stem cells (CSCs) [10,11,12]. In addition to being responsible for tumor growth, CSCs drive metastatic spread and chemoresistance in several cancer types including PDAC [13,14]. QM carcinoma cells may also assemble an immunosuppressive tumor microenvironment and unlike their E counterparts develop resistance to anti-CTLA4 immune checkpoint blockade therapy in part by cross-protection of their more E counterparts from immune attack [15]. 

Similar to QM tumor cell development, an EMT-like process is also involved in early pancreas development. During the stepwise trajectory of endocrine progenitor maturation, the delaminating cells are enriched in EMT signature markers and co-express ECAD and VIM [16]. Moreover, these cells express Neurogenin 3 (Ngn3), a key endocrine progenitor transcription factor necessary for endocrine cell specification [17], and endocrine precursor cell delamination through *Snai2* expression [18]. A growing and compelling line of evidence from a variety of models and experimental approaches in rodents and humans suggests that normal (or neoplastic) ductal or acinar pancreatic cells can be converted to insulin-producing cells [19,20]. These pancreatic cell interconversions may occur in response to stimuli such as β cell loss, pancreatic injury, inflammation, or metabolic stress [20] and are mediated via the emergence of Ngn3. Finally, when expanded in vitro in monolayer culture functional β cells purified from adult human islets lose the β-cell phenotype, undergo an EMT, and become highly proliferative mesenchymal cells that retain the potential for re-differentiation into insulin-producing cells [21,22]. 

The aggressive nature and poor prognosis of QM compared with E tumors requires not only a pre-therapeutic biomarker-based stratification of PDAC patients to guide subtype-specific therapy decisions [4] but also a more effective targeting of QM cells. Due to the mechanistic connection between EMT programs and the CSC phenotype, tumor cells with mesenchymal attributes are believed to be more plastic as evidenced by their multilineage differentiation potential [23,24,25,26]. Cancer cell plasticity can thus be exploited therapeutically by forcing the TD of EMT-derived cancer cells into highly differentiated or specialized cells. Specifically, in a proof-of-principle work, Ishay-Ronen et al. converted highly invasive and disseminating murine breast cancer cells into post-mitotic functional adipocytes leading to the repression of primary tumor invasion and metastasis formation [24,25]. Unfortunately, analogous studies with human pancreatic cancer cells for possible future TD therapy in PDAC patients are lacking. However, the QM PDAC cell line, PANC-1, has been successfully directed in vitro towards ductal-to-endocrine differentiation through activation of *NEUROG3* [20,27] or induction of a β cell-like phenotype [28,29,30,31,32,33,34,35,36]. The variety of different protocols employed to achieve trans-differentiation suggests that these cells are particularly plastic; however, whether this is a consequence of their QM signature remains elusive as equivalent data for other PDAC-derived cell lines are not available. Based on these assumptions we now further pursued the idea that E and QM cells differ in their cellular plasticity with respect to ductal-to-endocrine TD. Using a series of non-isogenic and isogenic PDAC cell lines with well-characterized EMT phenotypes and metastatic potential, we assessed their sensitivity toward endocrine TD induced by either an epigenetic drug or growth factor/cytokines. We expect to obtain initial clues from in vitro work as to whether EMT-based TD therapy of PDAC tumors carries the potential to be developed further for possible future clinical application. 

## 2. Material and Methods

### 2.1. Pancreatic Cell Lines and Primary Culture

The pancreatic cancer cell lines BxPC-3, CAPAN-1, CAPAN-2, COLO 357, PANC-1, and MIA PaCa-2 were obtained from H. Kalthoff (Kiel, Germany) and L3.6pl cells from C. J. Bruns (Cologne, Germany). The IMIM-PC-1 and PaTu 8988s cell lines were a gift from A. Menke (Giessen, Germany) originally supplied by F.X. Real (Barcelona, Spain). The immortalized human pancreatic ductal epithelial (HPDE) cell line, H6c7, was purchased from Kerafast/Biozol (Eching, Germany). All cell lines were grown at 37 °C in the presence of 5% CO_2_ and maintained in RPMI 1640 (Life Technologies, Darmstadt, Germany), except for IMIM-PC-1 and PaTu 8988s cells, which were maintained in Dulbecco’s modified Eagle’s medium (DMEM). Both basal media were supplemented with fetal bovine serum (FBS, 10%), L-glutamine (2 mM), 1% Penicillin-Streptomycin-Glutamine (PSG, Life Technologies, Life Technologies, Darmstadt, Germany) and 1% sodium pyruvate (Merck Millipore, Darmstadt, Germany) as specified in previous publications [37,38]. The human insulin-secreting insulinoma cell line NT-3 was used as a positive control for insulin mRNA expression and was maintained as described earlier [39,40].

The primary line (LüPanc1) was derived from a tumor of a 64-year-old male patient undergoing distal pancreatectomy for PDAC of the pancreatic tail at our department. Histopathological assessment of the resected tumor revealed a moderately differentiated ductal adenocarcinoma of the pancreatic tail (pT3, N1, L1, V1, Pn1, R0, G2). Tumor cells were isolated immediately after surgical resection of the tumor by mechanical and enzymatic dissociation of a representative specimen. After seven days of cultivation in DMEM supplemented with FBS (20%) and penicillin/streptomycin, the first adherent colonies of epithelial tumor cells appeared and were surrounded by fibroblasts. The number of fibroblasts was successively reduced by differential trypsinization. After reaching stable growth of the cultured tumor cells, fibroblasts were depleted by an immunomagnetic assay based on fibroblast-specific antigens (Anti-Fibroblast MicroBeads, human, #130-050-601, Miltenyi Biotec, Bergisch Gladbach, Germany). 

### 2.2. Quantitative Real-Time PCR

The procedure and the conditions for real-time quantitative RT-PCR (qPCR), which was performed on an I-cycler with IQ software (Bio-Rad, Munich, Germany) were published earlier [38]. All values for the genes of interest were normalized to those for GAPDH or TBP, and relative gene expression was calculated according to the 2^−^^ΔΔ^Ct method. Amplification primers were chosen to span exon-intron boundaries. Sequence information of primers for the pancreatic endocrine and β cell-specific genes can be found in Appendix A and earlier publications from our group [39,41].

### 2.3. Immunoblotting

Cells were lysed in either RIPA or PhosphoSafe buffer (Calbiochem/Merck Millipore, Darmstadt, Germany) at 70–90% confluency and at different time points during continuous culture. Equal amounts of cellular proteins were fractionated by SDS-PAGE, transferred to PVDF membrane, and immunoblotted as described in detail earlier [38]. The antibodies used were anti-E-cadherin (#610181) and anti-Rac1 (#610650), BD Transduction Laboratories (Heidelberg, Germany); anti-Rac1b, (#09-271), Merck Millipore; anti-Vimentin, anti-GAPDH (14C10), #2118, Cell Signaling Technology (CST, Frankfurt am Main, Germany); anti-β-actin, Sigma (Deisenhofen, Germany); anti-Claudin-4, Clone 3E2C1, #18-7341, Zymed Laboratories (South San Francisco, CA, USA); anti-Snail, #3895, CST; anti-HSP90 (F-8), #sc-13119, Santa Cruz Biotechnology (Heidelberg, Germany). The HRP-linked anti-rabbit, #7074, and anti-mouse, #7076, secondary antibodies were from CST. HRP-linked secondary antibodies and Amersham ECL Prime Detection Reagent (GE Healthcare, Munich, Germany) were used for chemiluminescent detection of proteins on a BioRad ChemiDoc XRS imaging system. This device also allowed for the densitometric quantification of signal intensities from underexposed autoradiographs. The original Western blot images can be found at Appendix A.

### 2.4. Trans-Differentiation of PDAC-Derived Cells into Pancreatic Endocrine or Insulin-Expressing Cells

TD along the endocrine path was carried out according to a protocol by Lefebvre et al. [27]. In brief, the PDAC lines were treated with the DNA methyltransferase inhibitor, 5′-aza-2′-deoxycytidine (5′-Aza) (Sigma) or vehicle (0.1% dimethylsulfoxide, DMSO) in normal growth medium for 3 days followed by another 3 days in medium without 5′-Aza or vehicle.

TD into insulin-expressing cells used three different protocols involving either FGF-b+transferrin [29] (P1), proinflammatory cytokines [20] (P2), or IGF-1, SCF and transferrin [32] (P3). PANC-1 can be induced to form clusters that subsequently differentiate into hormone-expressing islet-like cell aggregates [29]. Various PDAC cell lines were cultured in RPMI 1640 or DMEM supplemented with 10% FBS, 1% PSG and incubated in humidified 5% CO_2_ and 95% air at 37 °C. At ~70% confluence, medium was removed, cells were treated with 0.05% trypsin/EDTA to loosen but not to detach the cellular monolayer from their extracellular matrix, a step that required different periods of time (i.e., 1 min for PANC-1, 15 min for COLO 357, and 10 min for L3.6.pl). Subsequently, cells were incubated in serum-free medium supplemented with 0.1% bovine serum albumin (BSA), 1.1 µg/mL transferrin (Sigma) and 500 ng/mL recombinant human fibroblast growth factor-basic (FGF-b/FGF-2, Preprotech, Hamburg, Germany) for 120 h with refreshment of the differentiation medium after 48 and 96 h. Control cells were cultured in normal growth medium without exposure to trypsin. Modifications include the replacement of DMEM by RPMI 1640, the extension of TDC from 96 to 120 h (and 7 d for COLO 357 cells), the omission of human placental lactogen and the pre-incubation period with high-glucose (4.5 g/L) containing medium. Preliminary experiments revealed that exposure to high-glucose reduced TD efficiency, probably because high glucose stimulates secretion of TGF-β1 from pancreatic cancer cells [42], some of which, i.e., PANC-1, display high autocrine production [43]. TGF-β1 in turn interferes with β cell differentiation [44,45] and negatively regulates adipogenesis in breast cancer cells [24]. Stimulation of COLO 357 and L3.6pl cells with proinflammatory cytokines rhTNF-α (50 ng/mL), rhIL-1β (25 ng/mL) and rhIFN-γ (100 ng/mL) (all from Preprotech) was carried out as described previously [20]. Alternatively, cells were seeded in 12 well plates and cultured in DME/F12 medium (Invitrogen/Thermo Fisher Scientific, Dreieich, Germany) containing 1% BSA, 5.5 µg/mL transferrin 10 ng/mL rhIGF-1 and 50 ng/mL rhSCF (both from Preprotech) for a period of 3 days. The differentiation medium was changed once on day 2 [32]. All other steps were carried out according to published protocols.

### 2.5. Real-Time Cell Invasion Assay

The invasive potential of COLO 357 and L3.6pl cells was determined with the impedance-based xCELLigence^®^ RTCA technology (Agilent Technologies, Santa Clara, CA, USA, supplied by OLS, Bremen, Germany) as outlined in detail elsewhere [38,46] with minor modifications. The underside of the porous membrane of the CIM plate-16 was coated with 30 μL of a 1:1 (*v*/*v*) mixture of collagens I and IV to enhance adherence of the cells and hence signal intensities. A total of 80,000 cells were loaded per well. Changes in impedance were recorded every 15 min and processed with the RTCA software (version 2.0, Agilent Technologies, Santa Clara, CA, USA, supplied by OLS, Bremen, Germany).

### 2.6. Statistics 

Data from the qPCR and immunoblot analyses were analyzed using unpaired two-tailed Student’s *t* test or the Wilcoxon test. Statistical significance was calculated from at least three independent experiments. The *p* values less than 0.05 were considered statistically significant and were denoted by one asterisk. In some experiments higher levels of significance were indicated (**, *p* ≤ 0.01; ***, *p* ≤ 0.001).

## 3. Results

### 3.1. Epithelial/Mesenchymal Phenotyping of PDAC-Derived Cell Lines Based on ECAD and VIM Expression

Initially, we determined the EMT phenotypes of a battery of established human PDAC cell lines of known histomorphological differentiation grade (G1-G3) by assessing the expression of ECAD and VIM by immunoblotting. Like the benign human pancreatic ductal epithelial cell line HPDE6c7, CAPAN-2 (G1), and to a lesser extent BxPC-3 (G2), presented with high levels of ECAD, while levels in PANC-1 (G2-G3) were low (Figure 1A). Of note, the levels of VIM followed an inverse pattern (Figure 1A). Yet other cell lines, CAPAN-1 (G1), COLO 357 (G1-G2), and IMIM-PC-1 (G unknown) all present with high or intermediate levels of ECAD and either lack VIM completely (CAPAN-1, IMIM-PC-1, Figure 1B) or display only low levels (COLO 357). In contrast, the poorly differentiated (G3) QM cell lines PANC-1 and MIA PaCa-2 [47] (Figure 1B) both of which are highly metastatic [48,49], either express low levels (PANC-1) or lack (MIA PaCa-2) ECAD protein but express high levels of VIM (Figure 1B). Unlike the E cell lines both QM lines were negative for Claudin-4 (CLDN4), a tight junctional protein and potent inhibitor of invasion and a metastatic phenotype of pancreatic cancer cells [50]. Moreover, mRNA levels of ECAD and those of additional epithelial markers, CLDN7 and EpCAM/CD326 were hardly detectable in MIA PaCa-2, intermediate in PANC-1 and high in COLO 357 cells (Appendix A). We included a third QM line, PaTu 8988s (G3) [51], which resembled MIA PaCa-2 with respect to high VIM and almost undetectable ECAD protein (Appendix A). The extreme QM phenotype of PaTu 8988s cells was further evident from high abundance of the mesenchymal marker und EMT promoter, RAC1, and absence of its E-associated splice isoform, RAC1b [37] (Appendix A).

Recently, we established a primary cell line (LüPanc1) from a PDAC of the pancreatic tail. These cells preferentially grow as clusters of cobblestone-like cells with the majority of cells presenting with an epithelioid morphology (Appendix A) but the occasional appearance of cells exhibiting an elongated shape (Appendix A, arrows). The mixed morphology corresponded to an intermediate EMT phenotype, i.e., high levels of both epithelial (ECAD, CLDN4) and mesenchymal (VIM) markers (Figure 1B), consistent with G2 differentiation grade. Taken together, except for LüPanc1, we found an inverse expression pattern of ECAD and VIM with a high ECAD:VIM ratio being characteristic for E/classical subtype cells (CAPAN-1/-2, COLO 357, BxPC-3, IMIM-PC-1) and low ECAD:VIM ratio marking the QM phenotype (MIA PaCa-2, PANC-1, PaTu 8988s) and being predictive of a strong invasive and metastatic potential. 

### 3.2. Epithelial/Mesenchymal Phenotyping of the Paired Isogenic COLO 357 and L3.6pl Cell Lines

The PDAC cell lines analyzed above were established from different patients and thus have a different genetic background, particularly with respect to activation of tumor suppressor or oncogenes [52], which may or may not impact the EMT process. To rule out the possibility of PDAC cell specificity, we extended our studies using paired isogenic pancreatic cancer cell lines (COLO 357 and L3.6pl) that differ in their metastatic behavior. L3.6pl (pancreas to liver) had increased metastatic potential selected by repeated cycles of injection of the parental cells into the pancreas of nude mice followed by retrieval of hepatic metastases and reinjection into the pancreas [53,54]. Given the association of invasive/metastatic capacity with the QM phenotype, we hypothesized that L3.6pl cells should have higher or lower abundance of VIM or ECAD, respectively, than the parental cells. To this end, L3.6pl cells expressed more VIM but less ECAD and CLDN4 than COLO 357 cells (Figure 2A), suggesting that they had acquired a more mesenchymal phenotype during the cycles of in vivo selection. Finally, L3.6pl cells expressed lower transcript levels of the epithelial marker genes *KRT18*, *CLDN7*, *EPCAM* and *CDH1* (Figure 2B). These data show that L3.6pl cells have stably acquired a couple of mesenchymal traits as a result of adaptive selection for a metastatic phenotype.

### 3.3. The EMT Phenotype of PDAC Cells Determines Their Trans-Differentiation Potential to Pancreatic Endocrine Precursors 

Next, we subjected the PDAC-derived cell lines analyzed in Figure 1 to endocrine trans-differentiation culture (TDC) to assess their potential to convert into cells with a pancreatic endocrine phenotype. Initially, we adopted a protocol using treatment with the chemical demethylating agent, 5′-Aza [27], a DNA methyltransferase (DNMT) inhibitor. Subsequently, cells were assayed for transcriptional induction of *NEUROG3*, encoding NGN3, a master transcription factor in endocrine differentiation associated with immature or non-functional β/δ cells. Results show that the response of CAPAN-1, COLO 357, and IMIM-PC-1 cells was either weak or absent even at the higher (10 µM) concentration. In contrast, MIA PaCa-2, PANC-1, and PaTu 8988s cells responded strongly even to the low concentration (1 µM) of 5′-Aza (Figure 3). The primary PDAC cell line (LüPanc1) presented with a ~2 fold induction of *NEUROG3* at 1 µM 5′-Aza (Appendix A), the extent of which was comparable to that in the E cell lines. 

### 3.4. The EMT Phenotype of PDAC Cells Determines Their Trans-Differentiation Potential to Insulin-Expressing Cells

The QM cell line PANC-1 has the potential to be converted or trans-differentiated into insulin-producing cells but how this ability compares to other PDAC cells has not yet been studied. From the variety of different protocols [28,29,30,31,32,33,34,35,36] we chose that of Donadel et al. which involves a 5 d treatment with FGF-b and transferrin in serum-free medium [28]. Upon application of this protocol (designated P1) to PANC-1, MIA PaCa-2, COLO 357, and LüPanc1 cells we noted in PANC-1 and MIA PaCa-2 (Appendix A) but not in COLO 357 and LüPanc1 (not shown) the appearance of cell aggregates resembling pancreatic islets. Moreover, all four cell lines activated a β cell-specific transcriptional program as evidenced by upregulation of mRNA for *INS* and other markers of mature functional pancreatic β cells, such as the glucose transporter and metabolic sensing enzyme, GLUT2 (encoded by *SLC2A2*), and MAF BZIP Transcription Factor A (encoded by *MAFA*). MAFA is a transcription factor that binds a conserved enhancer element (RIPE3b) to regulate pancreatic β cell-specific expression of insulin [55] (Figure 4A). Of note, the QM cell lines PANC-1 and MIA PaCa-2 cells were clearly more responsive with respect to the induction of these three genes than E-type COLO 357 or intermediate-type LüPanc1 cells (Figure 4A). Within the QM class PANC-1 displayed a greater *INS* and *SLC2A2* response than MIA PaCa-2 cells (Figure 4A). The transdifferentiated PANC-1 and MIA PaCa-2 cells expressed 80.63% and 56.1%, respectively, of the INS mRNA levels seen in the insulin-producing human insulinoma cell line NT-3 [39,40], while the abundance of SLC2A2 and MAFA transcripts was higher and lower, respectively, than in NT-3 cells (Appendix A). Conversion to a pancreatic endocrine phenotype was further validated in COLO 357 cells by activation of other genes known to be involved in early endocrine development, i.e., *INSM1*, *NEUROG3*, *NEUROD1*, *NKX2-2*, *PAX4,* and *PAX6* [56,57] (Appendix A). Moreover, FGF-b/transferrin-based TDC or 5′-Aza treatment of PANC-1 cells resulted in concurrent changes in gene expression indicative of a mesenchymal-epithelial transition (MET), i.e., downregulation of SNAIL, RAC1 (Figure 4B), and VIM (Appendix A). This is consistent with the notion that PDAC cells are competent to undergo MET regardless of which program they use to shed their epithelial program during EMT [58]. To reveal if proliferation rates of PANC-1 and MIA PaCa-2 cells are affected by endocrine TD, we performed cell counting assays. To this end, application of P1 or an alternative protocol involving a combination of IFN-γ, IL-1β, and TNF-α in normal growth medium (designated P2) was associated with a strong decrease in cell numbers in both cell lines (Figure 4C). Of note, preincubating PANC-1 cells with the proinflammatory cytokine mix of P2 also strongly reduced their migratory activity (Figure 4D). Taken together with the results from Figure 3, we conclude that QM PDAC cell lines respond much stronger than well-differentiated E lines to epigenetic drug or FGF-b+transferrin-based pancreatic endocrine differentiation signals. In addition, endocrine TD is associated with partial reversal of mesenchymal differentiation and the cancer phenotype.

### 3.5. The Increased Metastatic Potential of L3.6pl Cells Compared with COLO 357 Cells Is Associated with a Higher Trans-Differentiation Potential to Insulin-Expressing Cells

Although the susceptibility to pancreatic endocrine differentiation appeared to be associated with the EMT phenotype among the various cell lines tested (low in E and high in QM cells), we sought to confirm this with cells on an isogenic background. Having shown above that L3.6pl cells more abundantly express VIM (see Figure 2) and less ECAD than COLO 357 cells [53]—suggesting that they were more mesenchymal—we wanted to know if this was reflected in an increased invasive potential in vitro. Intriguingly, L3.6pl cells migrated more vigorously in real-time cell migration assays than COLO 357 cells (Figure 5A) in accordance with their higher metastatic activity in vivo [53].

Next, we asked whether the metastatic isogenic variant is also more susceptible to ductal-to-endocrine TD. Given that chronic pancreatitis is a risk factor for PDAC and that an inflammatory environment can directly promote tumor initiation [42,59], we employed here TD protocols P2 and a third protocol (P3, based on a combination of IGF-1, SCF and transferrin) for the generation of insulin-expressing cells. In fact, a comparison revealed that the ductal-to-endocrine differentiation response of the L3.6pl cells to both TDC protocols was more pronounced than that of their parental counterpart as assessed by activation of *INS* and *SLC2A2* (Figure 5B). The *MAFA* gene was not consistently induced with these alternative TD protocols, suggesting that other transcription factors account for upregulation of insulin and GLUT2 mRNA. Together, these data extend those in Figure 3 and Figure 4 in that higher plasticity can result from a host microenvironment-induced shift in the EMT phenotype and was thus directly related to non-genetic changes associated with acquisition of metastatic competence. 

## 4. Discussion

PDAC is a treatment-refractory malignancy in urgent need of a molecular framework for guiding therapeutic strategies. Bulk transcriptomic efforts over the last decade have yielded two broad consensus subtypes, E/classical and QM/squamous/basal-like that correspond to a different extent of histomorphological de-differentiation. These subtypes, which correspond to different EMT phenotypes, are represented in the spectrum of permanent PDAC cell lines derived from either the patient’s primary tumor or a metastasis [48]. Based on ECAD:VIM expression ratios, CAPAN-1, CAPAN-2, COLO 357, IMIM-PC-1, and BxPC-3 cells were identified here as E and MIA PaCa-2, PANC-1, and PaTu 8988s cells as QM. PANC-1 cells share in common with the other two QM lines high VIM, but unlike these are ECAD low rather than null, consistent with moderate-to-poor histomorphological differentiation [48,60]. A primary cell line established in our lab from a PDAC located in the pancreatic tail had a mixed phenotype with VIM levels comparable with those of the QM cell lines but ECAD levels in the range of those from our E lines.

PANC-1 cells were successfully converted to insulin-producing cells using a variety of protocols involving either growth factors [29,32,36], agonists of proteinase-activated receptors [34], thyroid hormones [35], high-glucose [33], the diterpene lactone andrographolide [28], or forced expression of the *INS* promoter or INSM1, a zinc-finger transcription factor with restricted expression in β cells during early pancreas development and transiently activated by NGN3 [31]. Since PANC-1 was the only PDAC-derived cell line for which TD into insulin-producing cells has been reported, we next assessed in a comparative fashion the response of various other PDAC cell lines with either an E or QM signature to treatment with the demethylating drug, 5′-Aza, or FGF-b+transferrin with respect to activation of a pancreatic endocrine or pancreatic β cell-specific transcriptional program. We observed that the QM phenotype endowed PANC-1 and MIA PaCa-2 cells with a greater TD potential compared with the E phenotype of CAPAN-1, COLO 357, IMIM-PC-1, or the E/mixed phenotype of LüPanc1 cells as verified by the expression of various markers of endocrine development, β cell differentiation, or the mature β cell phenotype [39]. Re-differentiation of TDC-treated cells was also evident from an increase in the abundance of ECAD and a concomitant decrease in that of SNAIL protein. These results suggest that QM cells are more plastic than E cells with respect to ductal-to-endocrine TD.

Pancreatic islets of Langerhans originate from endocrine progenitor cells within the pancreatic ductal epithelium. Concomitant with differentiation of these progenitors into hormone-producing cells these cells delaminate, aggregate, and migrate away from the ductal epithelium. This process is governed by a partial EMT [16], suggesting that regulation of EMT and stemness/pluripotency gene expression may extend to pancreatic islet morphogenesis and islet cell migration. Ngn3-positive endocrine progenitor cells of the developing endocrine pancreas also express Slug, whose expression is maintained during endocrine cell differentiation where it becomes increasingly restricted to the insulin-producing β cells [18]. Human islet cell expansion in culture results in loss of the β-cell phenotype and induction of EMT along with activation of the ZEB1 and ZEB2 genes [21,22]. Conversely, downregulation of ZEB1 in expanded β cells induced an MET, β cell gene expression, and growth inhibition. In addition, inhibition of ZEB1 expression potentiated re-differentiation to a β cell-like phenotype in response to a combination of soluble factors [22]. Hence, transcription factors activated during EMT were permissive for β cell genesis, indicating that transcriptional regulation of EMT and MET is directly connected to the ductal-to-endocrine differentiation potential of cancer cells. Moreover, the acquisition of stem-like properties with the re-activation of stem pathways through a profound phenotypic EMT-based cell reprogramming in both normal pancreas development and pancreatic carcinoma progression appears as a significant cause of cellular plasticity. The importance of epigenetic events rather than alterations in tumor suppressor or oncogenes in gaining a plastic state is emphasized by the ability of the demethylating agent, 5′-Aza, to preferentially increase the TD potential of QM rather than E cells of ductal origin. This is supported by results from the paired COLO 357-L3.6pl isogenic model, in which non-genetic alterations acquired during in vivo selection of the parental COLO 357 cells towards a more metastatic phenotype also account for increased plasticity.

Among the three QM cell lines, PANC-1 turned out to be the most plastic upon 5′-Aza and growth factor-based TDC. The current view holds that cells with partial EMT are the most plastic ones. Interestingly, in contrast to MIA PaCa-2 and PaTu 8988s, PANC-1 cells exhibit a moderate rather than an extreme mesenchymal morphology and have retained some epithelial features, i.e., readily detectable levels of ECAD in immunoblots. Moreover, in contrast to MIA PaCa-2 [37] and PaTu 8988s (Appendix A) they express readily detectable levels of RAC1b, an alternatively spliced isoform of *RAC1* associated with the E subtype in both PDAC [37] and breast cancer [46] cells. Using a lineage-labeled mouse model of PDAC to study EMT in vivo, it was found that QM subtype cells utilize transcriptionally dominated programs to lose their epithelial phenotype during EMT (designated complete EMT), while those belonging to the E subtype rely on protein re-localization (referred to as partial EMT) [58]. Of note, although both MIA PaCa-2 and PANC-1 cells were predicted cells with complete EMT, PANC-1 was more closely related to cells with partial EMT (and hence the E subtype) than MIA PaCa-2 (see Figure 6B in Ref. [58]).

Phenotypic reprogramming accompanied by SC formation may have a role in pancreatic disease pathology. For instance, in a diabetic setting, chronic exposure of pancreatic epithelial cells to high-glucose can result in activation of EMT by TGF-β, with a subsequent increase in SC properties [42]. Chronic pancreatitis, a known risk factor for PDAC is known for transient pulses of TGF-β that could convert E cells to QM cells and contribute together with pre-existing genetic changes to PDAC development [61]. TGF-β1-induced EMT increases the number of CSCs in PANC-1 cells [62,63] and TGF-β1, along with TNF-α, synergistically enhances the CSC properties of MIA PaCa-2 cells [64]. Moreover, high local concentrations of inflammatory mediators in the tumor microenvironment may facilitate cancer progression by directly activating EMT programs, eventually leading to mesenchymal conversion and invasion/metastasis [65]. Moreover, localized inflammation can trigger the awakening of dormant cells and their development into macroscopic metastases, and this escape from latency is dependent on the expression of Zeb1 [66]. Our finding that cells established from a lymph node/liver metastasis—COLO 357/L3.6pl—can be induced by a cocktail of inflammatory cytokines to activate a β cell-specific transcriptional program in vitro rather than undergoing inflammation-induced EMT [65] is intriguing in view of a potential future in vivo application of a TD therapy.

The immunosuppressive and aggressive/metastatic state of a tumor predominated by QM cancer cells also offers a promising and highly innovative therapeutic option—exploiting the increased plasticity of QM cells to enforce their TD into highly specialized growth-arrested cells. Using a well-established adipogenesis induction protocol, Ishay-Ronen et al. have provided proof-of-principle that murine mesenchymal-like breast cancer cells can be converted to terminally differentiated post-mitotic adipocytes along with repression of primary tumor invasion and metastasis formation in mouse models of breast cancer [24,25]. At the tumor-host interface, the invasive front and tumor buds provide evidence of partial EMT induction and are suspected to be the main location of invasive cancer cells with increased cell plasticity [67,68]. Of note, targeting these cells with adipogenic TD therapy ablated the highly invasive tumor fronts of treated mice and inhibited tumor invasion, suggesting that TD may be particularly effective in preventing the very early stages of tumor cell dissemination. As cellular models of EMT-induced cancer cell plasticity Ishay-Ronen et al. employed mammary epithelial cells that had undergone either a complete but reversible EMT (induced by long-term treatment with TGF-β in vitro), or an irreversible EMT (induced by engineered ablation of *CDH1*) [24]. It will be interesting to see how prior mesenchymal conversion of human E PDAC lines (i.e., CAPAN-2, COLO 357, and IMIM-PC-1, all of which are TGF-β-sensitive) with TGF-β impacts plasticity and the potential for ductal-to-endocrine TD. In addition, the reversibility of the TD process needs to be assessed, i.e., whether differentiated PDAC cells maintain their β cell characteristics or revert to a mesenchymal state upon removal of the TD inducers. Finally, we observed that the P1 and P2 TD methods impacted the cancer phenotype in that both cell counts and migration rates were reduced. Although the expression profiles of proliferation and invasion-associated genes during endocrine TD still need to be determined, our data so far suggest that transdifferentiated cells become less proliferative and less invasive.

A TD approach of highly metastatic PDAC cells into mature β cells with pancreatic endocrine or β cell differentiation agent(s) in combination with a TGF-β inhibitor [44,45] could be an attractive therapeutic option for PDAC patients. If the resulting β cell-like cells are capable of glucose-regulated insulin secretion, they might be able to compensate in pancreatic cancer patients the insulin deficiency resulting from radical surgery or a long-term history of type II diabetes. This “two birds with one stone” strategy is even more appealing than the “better fat than dead” approach of adipocytic conversion of aggressive breast cancer cells [69]. When applied to PDAC patients the latter strategy might pose risks due to de-differentiation of adipocytes to cancer-associated adipocytes, which have been shown to promote pancreatic cancer progression [70]. In sum, the results of this study have yielded initial clues that a de-differentiation process, which enhances cellular plasticity may be exploited therapeutically to enforce the TD of EMT-derived pancreatic cancer cells into non-malignant functional insulin-expressing cells. 

## 5. Conclusions

Pancreatic cancer cells of ductal origin and an invasive/metastatic quasi-mesenchymal (QM) signature can be induced to a greater extent than their epithelial (E) counterparts to differentiation into β cell-like cells, as assessed by the activation of the insulin gene and other early endocrine and β cell-specific marker genes. Our results underscore the notion that the QM phenotype coincides with increased cell plasticity and heightened trans-differentiation potential, suggesting that pancreatic duct cancer cells undergoing EMT gain the potential to activate a pancreatic β cell-specific transcriptional program. Follow-up studies with mouse models need to evaluate whether indeed QM cancer cells can be converted into bona fide β cells in vivo and whether this is facilitated by pharmacologic inhibition of signaling pathways that interfere with β cell differentiation. Since QM tumors have been demonstrated to exhibit increased resistance to standard chemo- and immunotherapy, we predict that the removal of invasive mesenchymal cancer cells by trans-differentiation therapy may also overcome therapy resistance and cancer relapse.

## Figures and Tables

**Figure 1 cancers-13-04663-f001:**
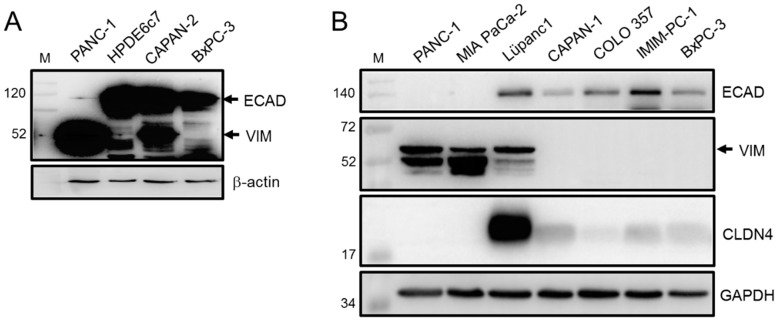
Expression of epithelial and mesenchymal markers in PDAC-derived tumor cell lines as measured by immunoblot analysis. (**A**) The indicated PDAC cell lines, and HPDE6c7 cells as control, were immunoblotted for ECAD and VIM. Equal loading was verified by detection of β-actin. The blot was overexposed to also catch the weak ECAD band in PANC-1 and the lack of VIM signals in HPDE6c7 and BxPC-3 cells. For a short exposure copy of this blot see Appendix A. (**B**) As in (**A**) except that additional PDAC cell lines were analyzed for ECAD, VIM, CLDN4. Detection of GAPDH served as a loading control. In lysates from COLO 357 cells a band for VIM becomes visible upon longer exposure of the blot (not shown).

**Figure 2 cancers-13-04663-f002:**
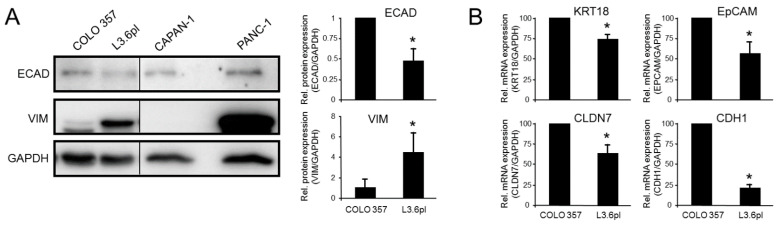
EMT phenotyping of COLO 357 and L3.6pl cells. (**A**) COLO 357 and L3.6pl cells were studied by immunoblotting for ECAD and VIM. GAPDH was used as a loading control. PANC-1 cells served as a positive and CAPAN-1 cells as a negative control for VIM expression. The graphs to the right represent data quantification of VIM and ECAD (mean ± SD of 3–6 samples taken at regular intervals during continuous culture). The asterisks indicate significant differences (*p* < 0.05, Wilcoxon test). (**B**) The same two cell lines were subjected to qPCR analysis of *KRT18*, *CLDN7*, *EPCAM* and *CDH1*. Data represent the mean ± SD of three parallel wells. The assay shown is representative of three assays performed in total. The asterisks (*) denote significance (*p* < 0.05, unpaired two-tailed Student’s *t* test).

**Figure 3 cancers-13-04663-f003:**
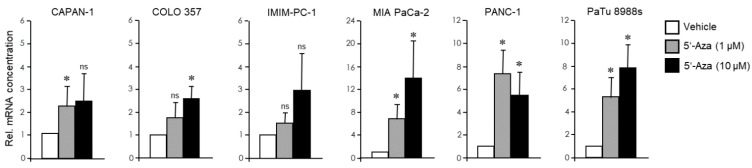
Effect of treatment with 5′-Aza on *NEUROG3* activation in PDAC-derived tumor cells with E or QM signatures. The indicated PDAC cell lines were treated with 5′-Aza (1 or 10 µM), or vehicle, in normal growth medium for 3 d followed by another 3 day incubation period in medium without 5′-Aza. NGN3 expression was analyzed by qPCR. Please note the different scales of the graphs when comparing the 5′-Aza effects among cell lines. Data are plotted relative to vehicle controls and represent the mean ± SD of three (IMIM-PC-1, PANC-1, PaTu 8988s) or four (CAPAN-1, COLO 357, MIA PaCa-2) independent experiments and are plotted relative to control cells set arbitrarily at 1.0. Asterisks (*) indicate significance (*p* < 0.05, Wilcoxon test); ns, non-significant.

**Figure 4 cancers-13-04663-f004:**
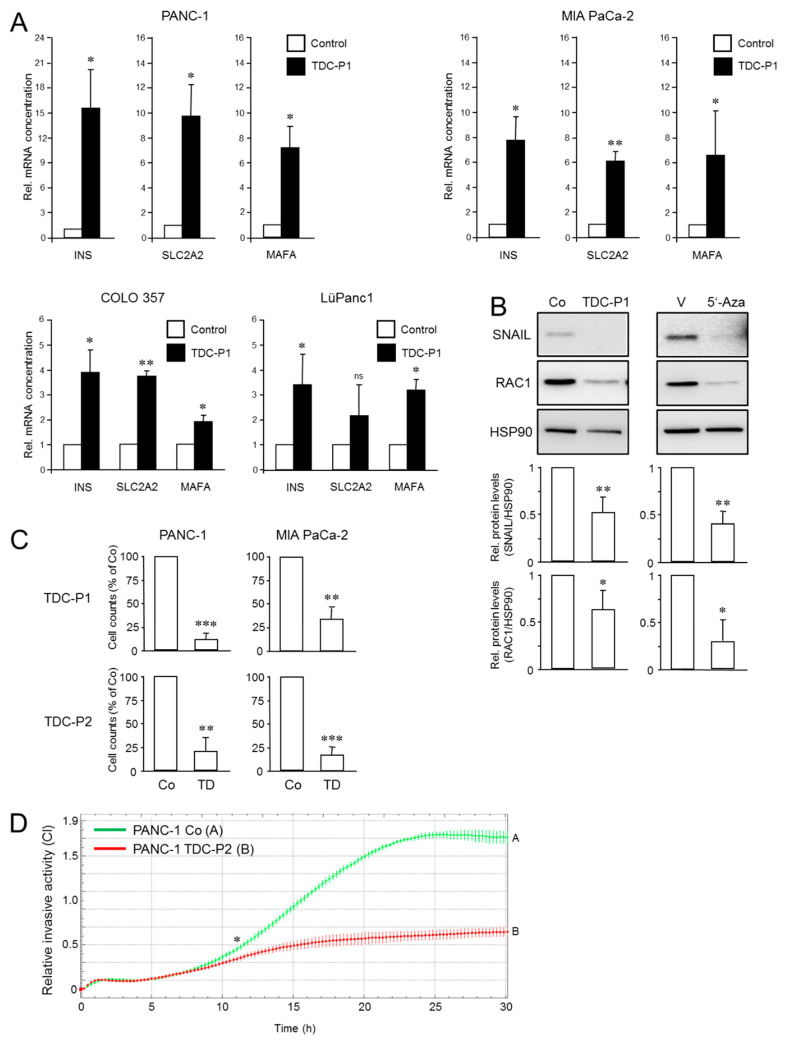
In vitro TD of established and primary pancreatic ductal epithelial cell lines to β cell-like cells. (**A**) PANC-1, MIA PaCa-2, COLO 357, and LüPanc1 cells underwent TD with FGF-b and transferrin (TDC-P1) for 5 d and after RNA isolation were monitored by qPCR for the induction of genes that specify mature functional β cells (*INS*, *SLC2A2*, *MAFA*). The induction of *SLC2A2* in LüPanc1 cells missed statistical significance (ns). Data represent the mean ± SD of three independent experiments (*n* = 3). Asterisk, * *p* < 0.05, Wilcoxon test. Please note the different scales on the ordinates when comparing mRNA levels among the various cell lines. (**B**) Immunoblot analysis of SNAIL and RAC1 in protein lysates sampled from PANC-1 cells at the end of TDC-P1 or treatment with 5′-Aza. Detection of HSP90 served to verify equal gel loading. The graphs underneath the blots show results from densitometry-based quantification of band intensities (mean ± SD, *n* = 3). Co, control culture. (**C**) Effect of ductal-to-endocrine TD on cell proliferation. PANC-1 and MIA PaCa-2 cells first underwent TDC with either P1 (for 3 d rather than the usual 5 d to avoid density-mediated growth arrest of control cells) or P2 followed by manual cell counting using a Neubauer chamber. Data (mean ± SD of triplicate wells) are representative of three assays and are displayed relative to (non-transdifferentiated) Co cells. (**D**) Effect of ductal-to-endocrine TD on cell migration. PANC-1 cells were subjected to real-time cell migration assay using xCELLigence technology following TDC-P2 for 3 d. The assay itself was performed in normal growth medium in the absence of added proinflammatory cytokines. Data are the mean ± SD of quadruplicate wells and values for transdifferentiated cells are significantly different from controls at 11:00 (denoted by an asterisk (*)) and all later time points. The assay shown is representative of three assays performed in total. ** *p* ≤ 0.01; *** *p* ≤ 0.001.

**Figure 5 cancers-13-04663-f005:**
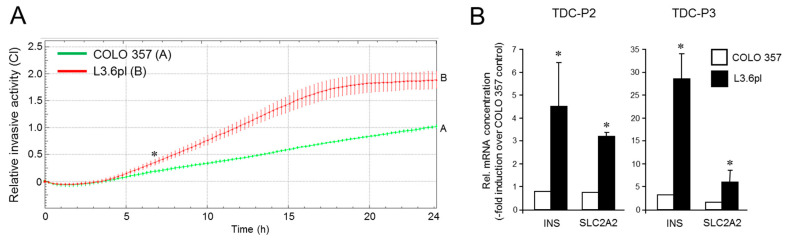
Comparison of the invasive and β cell TD potential of COLO 357 and L3.6pl cells. (**A**) Real-time cell migration assay with COLO 357 and L3.6pl cells. Data are the mean ± SD of quadruplicate wells and are significantly different between cell lines at 6:45 (denoted by an asterisk) and all later time points. The assay shown is representative of three independent RTCA assays. (**B**) COLO 357 and L3.6pl cells were cultured with P2 (TDC-P2), or a combination of IGF-1, SCF and transferrin (TDC-P3), for 72 h and assayed for *INS* and *SLC2A2* mRNA abundance by qPCR. The data shown are from a representative experiment out of three experiments performed in total (mean ± SD of triplicate wells). Displayed are the values for TDC in relation to those of COLO 357 cells with control medium. The asterisks (*) denote significance (*p* < 0.05, unpaired two-tailed Student’s *t* test). The changes in INS and SLC2A2 RNA expression between COLO 357 cells treated with either TDC-2 or TDC-3 and the respective untreated controls were also significant (not shown for reasons of clarity).

## Data Availability

Data is contained within the article or Appendix A.

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
