# Peer review of "A Comparative Endocrine Trans-Differentiation Approach to Pancreatic Ductal Adenocarcinoma Cells with Different EMT Phenotypes Identifies Quasi-Mesenchymal Tumor Cells as Those with Highest Plasticity"

_cancers, 2021, doi:10.3390/cancers13184663_

Round 1

Reviewer 1 Report

The manuscript titled "A comparative endocrine trans-differentiation approach to pancreatic ductal adenocarcinoma cells with different EMT phenotypes identifies quasi-mesenchymal tumor cells as those with highest plasticity", compared the trans-differentiation into non-malignant pancreatic or non-pancreatic cells, of both epithelial (E) or quasi-mesenchymal (QM) PDAC-derived cell lines. The results obtained showed as QM cells more strongly responded to the differentiation compared to E cell lines. opening novel therapeutic strategy to convert metastatic PDAC cells into benign post-mitotic cells. The argument is very interesting and the paper is well written in each section. I have few points that need to be addressed before this paper is considered for publication.

Specific comments:

  • In the abstract the literature gap is missed;
  • Have the authors also tested some other markers of EMT, like MMP9?
  • For the trans-differentiation experiments, a positive control, like MIN6 cells, should be included
  • Have the authors tested the EMT markers following trans-differentiation? This is a key point to asses the differentiation of PDAC cells into post-mitotic benign cells.

Author Response

Dear Editor, dear Anica:

This letter of submission is accompanied by our revised manuscript entitled:

“A comparative endocrine trans-differentiation approach to pancreatic ductal adenocarcinoma cells with different EMT phenotypes identifies quasi-mesenchymal tumor cells as those with highest plasticity”

We are indebted to the reviewers for their valuable comments and suggestions and have done our best to incorporate these into the revised version of our manuscript (highlighted in the “track changes” mode).

We believe that the reviewers’ critiques have substantially improved the quality of our manuscript and we are looking forward to its final acceptance in Cancers.

Sincerely yours,

Hendrik Ungefroren

Reviewer1

The manuscript titled "A comparative endocrine trans-differentiation approach to pancreatic ductal adenocarcinoma cells with different EMT phenotypes identifies quasi-mesenchymal tumor cells as those with highest plasticity", compared the trans-differentiation into non-malignant pancreatic or non-pancreatic cells, of both epithelial (E) or quasi-mesenchymal (QM) PDAC-derived cell lines. The results obtained showed as QM cells more strongly responded to the differentiation compared to E cell lines. opening novel therapeutic strategy to convert metastatic PDAC cells into benign post-mitotic cells. The argument is very interesting and the paper is well written in each section. I have few points that need to be addressed before this paper is considered for publication.

Specific comments:

  • In the abstract the literature gap is missed;

Response: This piece of information has been added.

  • Have the authors also tested some other markers of EMT, like MMP9?

Response: As shown in Fig. 2B, expression of the epithelial markers KRT18, CLDN7, EpCAM and CDH1 was measured in COLO 357 and L3.6pl cells by qPCR (Fig. 2B). In the course of the revision, we have also compared expression levels of CLDN7, EpCAM and CDH1 in MIA PaCa-2, PANC-1 and COLO 357 cells by qPCR. We found that the mRNA levels of all three genes were higher in PANC-1 than in MIA PaCa-2 but lower than in COLO 357 cells and for CDH1 these correspond nicely with the ECAD protein levels. We have included these qPCR data as new Table S2 in the Supplementary material. Data on MMP9 expression in PANC-1 cells have been published previously (Ref. #38 in the revised version).

  • For the trans-differentiation experiments, a positive control, like MIN6 cells, should be included

Response: As requested, we have used as positive control the insulin-secreting human insulinoma cell line NT-3 (References #39 and #40 in the revised version) since MIN6 cells were not available to us. Data show that transdifferentiated PANC-1 and MIA PaCa-2 cells expressed 80.63 % and 56,1 %, respectively, of the INS mRNA levels seen in NT-3 cells, while the abundance of SLC2A2 and MAFA transcripts was higher and lower, respectively, than in NT-3 cells. These data have been added to the Supplementary material as Figure S3B.

  • Have the authors tested the EMT markers following trans-differentiation? This is a key point to asses the differentiation of PDAC cells into post-mitotic benign cells.

Response: Yes, this shown in Figure 4B and Figure S4. Here, we have tested Snail, Rac1 and VIM and observed a downregulation upon with FGF-b+transferrin (now designated protocol 1 (P1) in the revised version) and 5’-Aza, respectively. However, we were unable to observe concurrent upregulation of E-cadherin. In response to a request from Reviewer 3, we have analysed additional EMT-related cellular responses such as proliferation and cell motility in PANC-1 and MIA PaCa-2 cells following TDC-P1 or TDC with IFN-γ+IL1-β+TNF-a (designated P2 in the revised version, see new figure panels 4C+D). These new data show that these cells become less proliferative and less invasive.

Reviewer 2 Report

This is a paper of the quasi-mesenchymal subtype of pancreatic ductal adenocarcinoma. The data may certainly be of use for treatments in the future. It will be easier to understand if you add a supplement figure S1-4 in the text.

Author Response

Dear Editor, dear Anica:

This letter of submission is accompanied by our revised manuscript entitled:

“A comparative endocrine trans-differentiation approach to pancreatic ductal adenocarcinoma cells with different EMT phenotypes identifies quasi-mesenchymal tumor cells as those with highest plasticity”

We are indebted to the reviewers for their valuable comments and suggestions and have done our best to incorporate these into the revised version of our manuscript (highlighted in the “track changes” mode).

We believe that the reviewers’ critiques have substantially improved the quality of our manuscript and we are looking forward to its final acceptance in Cancers.

Sincerely yours,

Hendrik Ungefroren

Reviewer 2

This is a paper of the quasi-mesenchymal subtype of pancreatic ductal adenocarcinoma. The data may certainly be of use for treatments in the future. It will be easier to understand if you add a supplement figure S1-4 in the text.

Response: The data in the supplementary figures are not mandatory for the understanding or the conclusions and will be easily available online for the reader. For this reason and because additional data requested by the other two reviewers were included in the main body of the manuscript, we would prefer to leave them in the Suppl. material and not include them in the main body of the manuscript.

Reviewer 3 Report

The study aims to determine the differentiation potential of diverse PDAC cell lines. The results indicate the higher trans-differentiation potential of QM phenotype in various in vitro studies.

Overall, this is an original approach to treat highly aggressive PDAC and could become relevant to better understanding of cancer cell plasticity and its therapeutic potential. The study has its focus on important and relevant questions in the field of EMT and cell plasticity and can support the possibility to utilize trans-differentiation therapies in aggressive tumors.

The outlined methodology and general overview of the results support the suggested hypothesis. Yet, the interpretation of the results is in some cases inconsistent.  This is a major concern in figure 4 and the corresponding text as detailed hereafter.

Moreover, while the results indicate beta-cells differentiation with appropriate RNA up-regulation, the interpretation should be supported with more extensive methodology e.g. protein-based analysis. It is also not demonstrated how PDAC-related gene expression, proliferation rate, and/or migration are affected by the trans-differentiation method. Concluding that this study suggests a novel therapeutic approach requires more extensive analysis to demonstrate the effect on cancer phenotype.

Of note, since different trans-differentiation methods are used, a clearer presentation in the figure legends but also in the method section would be beneficial.

Specific comments:

Figure 1A- A cleaner WB is required.

Figure 1B- Vimentin lane is cut too close to the bars

Figure S1B– Some of the arrows seem to point at large, perhaps cells in senescence, rather than elongated morphology

Figure 3– Capan-1 cells demonstrated the same statistical increase in NEUROG3 expression as the QM cells. Also, please note in the figure legend that the results indicate NEUROG3 up-regulation rather than activation.

Figure 4A and 323-325, 335-338: The interpretation disagrees with the results presented. Namely, graph shows similar upregulation in COLO-357 compared to QM cells.

Figure 5B– Please state if COLO-357 demonstrate significant change in RNA expression

20-21 – The paper does not demonstrate any trans-differentiation into non-pancreatic cells

45- beta cells are proliferative endocrine cells and are not considered post-mitotic

68-EMT is not a genetic program

70 – EMT TF include also ZEB and twist, particularly important in PDAC

83-87 – additional references to support this part are required

285-286 – The link between the two parts of the sentence is not clear.

Author Response

Dear Editor, dear Anica:

This letter of submission is accompanied by our revised manuscript entitled:

“A comparative endocrine trans-differentiation approach to pancreatic ductal adenocarcinoma cells with different EMT phenotypes identifies quasi-mesenchymal tumor cells as those with highest plasticity”

We are indebted to the reviewers for their valuable comments and suggestions and have done our best to incorporate these into the revised version of our manuscript (highlighted in the “track changes” mode).

We believe that the reviewers’ critiques have substantially improved the quality of our manuscript and we are looking forward to its final acceptance in Cancers.

Sincerely yours,

Hendrik Ungefroren

Reviewer 3

The study aims to determine the differentiation potential of diverse PDAC cell lines. The results indicate the higher trans-differentiation potential of QM phenotype in various in vitro studies.

Overall, this is an original approach to treat highly aggressive PDAC and could become relevant to better understanding of cancer cell plasticity and its therapeutic potential. The study has its focus on important and relevant questions in the field of EMT and cell plasticity and can support the possibility to utilize trans-differentiation therapies in aggressive tumors.

The outlined methodology and general overview of the results support the suggested hypothesis. Yet, the interpretation of the results is in some cases inconsistent.  This is a major concern in figure 4 and the corresponding text as detailed hereafter.

Moreover, while the results indicate beta-cells differentiation with appropriate RNA up-regulation, the interpretation should be supported with more extensive methodology e.g. protein-based analysis. It is also not demonstrated how PDAC-related gene expression, proliferation rate, and/or migration are affected by the trans-differentiation method. Concluding that this study suggests a novel therapeutic approach requires more extensive analysis to demonstrate the effect on cancer phenotype.

Response: We have attempted to measure insulin secretion in transdifferentiated PANC-1 cells by ELISA but values were extremely low and could not be safely quantified. Rather than capture insulin protein production our intention was to demonstrate activation of a transcriptional program for pancreatic beta cells. For this reason, we have measured the induction of several genes involved in early endocrine differentiation and beta cell function rather than assessing only INS or only NEUROG3 as in most publications (i.e., Ref. #20).

As requested, we have measured proliferation rates in PANC-1 and MIA PaCa-2 cells in response to the FGF-b+transferrin and proinflammatory cytokine-based protocols (shown in the new panel C of Figure 4). The published protocols all required seeding of a relatively high cell number per surface area that rapidly resulted in growth arrest in the control cultures before completion of the 5-day transdifferentiation period with FGF-b+transferrin. For this reason, we have measured cell counts in these cultures already on day 3 when control cells were still sub-confluent rather than on day 5, the time of harvest for qPCR analysis of beta cell markers. This piece of information has been added to the legend of Figure 4.

In addition, we have determined the impact of the proinflammatory cytokine-based trans-differentiation protocol on cell migration and found that pre-incubation with this mix rendered PANC-1 cells less migration-active. A representative assay is shown in the revised version in the new panel D of Figure 4. Unfortunately, MIA PaCa-2 cells did not show migratory activity in the standard setup of the xCELLigence platform.

Of note, since different trans-differentiation methods are used, a clearer presentation in the figure legends but also in the method section would be beneficial.

Response: As requested, we have enhanced clarity with respect to the different trans-differentiation methods used. Specifically, the FGF-b+transferrin protocol has been designated P1, the inflammatory cytokine mix protocol P2 and the IGF-1+SCF+transferrin protocol P3. In the graph legends these three transdifferentiation protocols were denoted now as TDC-P1, TDC-P2 and TDC-P3, respectively.

Specific comments:

Figure 1A- A cleaner WB is required.

Response: This blot was intentionally overexposed to demonstrate the strong difference in E-cadherin and Vimentin expression between PANC-1 cells on the one hand and premalignant HPDE6c7 and epithelial PDAC cells on the other hand and the lack of Vimentin protein in HPDE6c7 and BxPC3 cells. However, we have added a short-exposure copy of this blot to the Supplementary material as new Figure S1A as noted in the respective figure legend. The former Figs. S1A and S1B became Figs. S1B and S1C, respectively.

Figure 1B- Vimentin lane is cut too close to the bars

Response: As requested, we have replaced the VIM blot with a copy that shows more space below the Vimentin bands. The upper band of the triplet represents the specific signal for vimentin (54 kD in size).

Figure S1B– Some of the arrows seem to point at large, perhaps cells in senescence, rather than elongated morphology

Response: This is a good point and may in fact be the case. Since a more detailed analysis of this issue is beyond the scope of this study, we have added a sentence to the legend of Figure S1 that is speculating on this possibility. It is definitely worthwhile to test this in future studies, i.e., by staining the cells for the senescence markers like b-gal.

Figure 3– Capan-1 cells demonstrated the same statistical increase in NEUROG3 expression as the QM cells. Also, please note in the figure legend that the results indicate NEUROG3 up-regulation rather than activation.

Response: It may have escaped the reviewer’s attention that the scales of the ordinate are different. At the 1 µM 5’-Aza concentration fold induction for Capan-1 is only 2.2, while for MIA PaCa-2, PANC-1 and PaTu 8898s cells it is ~7, 7.5 and 5.5-fold, respectively. We have decided to keep the graphs for the different cell lines at the same sizes for a better and uniform design. This required, of course, different scales on the ordinate (as stated in the legend). We apologize if this lack of clarity has caused confusion.

As requested, the term “activation” in the corresponding figure legend has been replaced by “upregulation”.

Figure 4A and 323-325, 335-338: The interpretation disagrees with the results presented. Namely, graph shows similar upregulation in COLO-357 compared to QM cells.

Response: It may have escaped the reviewer’s attention that the scales of the ordinates are different. For COLO-357, induction values are less than 4-fold for all three genes, while for PANC-1 and MIA PaCa-2 cells they range between 6 and 16-fold. Please compare relative numbers on the ordinates. Again, we are sorry for not having made this clear enough. To make sure that the reader becomes aware of this, we have added a short note to the legend (as in the legend of Figure 3).

Figure 5B– Please state if COLO-357 demonstrate significant change in RNA expression

Response: Changes in INS and SLC2A2 mRNA levels between COLO 357 cells treated with either TDC-P2 or TDC-P3 and the respective controls were statistically significant. This statement has been added to the legend of Figure 5 as “data not shown”.

20-21 – The paper does not demonstrate any trans-differentiation into non-pancreatic cells

Response: This is correct. We have deleted the term “non-pancreatic” from this line.

45- beta cells are proliferative endocrine cells and are not considered post-mitotic

Response: This is correct and we have therefore eliminated the term “post-mitotic” from this line.

68-EMT is not a genetic program

Response: This is correct and we have replaced the term “genetic” by “developmental”.

70 – EMT TF include also ZEB and twist, particularly important in PDAC

Response: We have added this sentence along with a reference (#9).

83-87 – additional references to support this part are required

Response: As requested, additional references have been added (#19, #20).

285-286 – The link between the two parts of the sentence is not clear.

Response: This sentence has been deleted since the mention of cancer stem cells is not directly relevant here.

Additional changes made:

  1. The two graphs in Figure 5B (now designated TDC-P2 and TDC-P3) have been switched to correspond with the description in the text.
  2. Three new references have been added and one has been deleted (Ref. #54 in the original version).

Round 2

Reviewer 3 Report

While the authors replied most of the comments and concerns, a major issue remains. The authors demonstrate a shift in gene expression towards pancreatic cells but this does not mean differentiation or trans-differentiation. Demonstration of differentiation requires protein quantification along with functional studies. These were not shown and in their reply the authors state that these results could not be achieved. This only means that there is still a way to go before trans-differentiation can be described. The manuscript could be published in my opinion but only if trans-differentiation or differentiation will be changed throughout the manuscript including title, abstract, figures and text. The correct interpretation would be gene expression shift towards pancreatic cells. This would be both exciting and correct.